# Study on Multi-Scale Coupled Ecological Dispatching Model Based on the Decomposition-Coordination Principle

**Tao Zhou \*, Zengchuan Dong, Wenzhuo Wang, Rensheng Shi, Xiaoqi Gao and Zhihong Huang**

Hydrology and Water Resources College, Hohai University, Nanjing 210098, China
\* Correspondence: 160401010016@hhu.edu.cn

**Abstract:** Studies on environmental flow have developed into a flow management strategy that includes flow magnitude, duration, frequency, and timing from a flat line minimum flow requirement. Furthermore, it has been suggested that the degree of hydrologic alteration be employed as an evaluation method of river ecological health. However, few studies have used it as an objective function of the deterministic reservoir optimal dispatching model. In this work, a multi-scale coupled ecological dispatching model was built, based on the decomposition-coordination principle, and considers multi-scale features of ecological water demand. It is composed of both small-scale model and large-scale model components. The small-scale model uses a daily scale and is formulated to minimize the degree of hydrologic alteration. The large-scale model uses a monthly scale and is formulated to minimize the uneven distribution of water resources. In order to avoid dimensionality, the decomposition coordination algorithm is utilized for the coordination among subsystems; and the adaptive genetic algorithm (AGA) is utilized for the solution of subsystems. The entire model—which is in effect a large, complex system—was divided into several subsystems by time and space. The subsystems, which include large-scale and small-scale subsystems, were correlated by coordinating variables. The lower reaches of the Yellow River were selected as the study area. The calculation results show that the degree of hydrologic alteration of small-scale ecological flow regimes and the daily stream flow can be obtained by the model. Furthermore, the model demonstrates the impact of considering the degree of hydrologic alteration on the reliability of water supply. Thus, we conclude that the operation rules extracted from the calculation results of the model contain more serviceable information than that provided by other models thus far. However, model optimization results were compared with results from the POF approach and current scheduling. The comparison shows that further reduction in hydrologic alteration is possible and there are still inherent limitations within the model that need to be resolved.

**Keywords:** ecological operation; multi-scale; decomposition-coordination; hydrologic alterations

## 1. Introduction

Reservoir ecological operation has become increasingly more significant and has recently drawn a lot of attention from research scientists in numerous scientific disciplines. By 2005, there were >45,000 reservoir dams higher than 15 m throughout the world, which are collectively capable of storing ~15% of annual global runoff [1]. While these reservoirs make a sizeable contribution to developing the economy, they are also the source of numerous ecological problems. The ultimate goal of ecological operation is to integrate ecological factors into standard operation objectives, such that ecological protection is given the same consideration as increasing profit margins. Essentially, the target of ecological operation is to mediate the conflict between society's demand and the river ecosystem's

needs [2]. As early as 1971, Schlueter put forward that reservoir management should maintain river diversity [3]. Following that pronouncement, a number of river ecological water demand calculation methods were developed—i.e., the Tennant Method, Wetted Perimeter Method, and IFIM Method, etc. [4]. Since then, increasing discharge from reservoirs to reserve enough ecological flow has become a common method of ecological operation [5–7].

Progressively more in-depth research on the riverine ecosystems has widened our understanding of the river system itself and the ecosystem it supplies. Thus, river ecological water demand no longer exclusively concerns the reserved water downstream. Many scholars have reached a broad consensus that riverine ecosystems adapt to the natural river flow pattern [8], and every river has its own characteristic flow pattern and corresponding biological community [9]. Thus, any flow pattern change—such as flood runoff, flow velocity, arrival time, and flood pulse frequency, etc., can cause a series of ecological consequences. As a result of these insights, ecological water requirements have been transformed from merely considering reserved flow [10,11] to considering the full extent of natural flow to which riverine biological communities must adapt [11–13]. Based on the above consensus, Richter et al., 1997 suggested implementing the Range of Variability Approach (RVA), which uses the Indicator of Hydrologic Alteration (IHA) to quantify how much the reservoir operation alters the riverine ecosystem. Homa et al. [11] presented the concept of ecological deficit and proposed minimizing ecological deficit as an objective of reservoir operation. Richter [14] offered a definition of sustainable water management called "Sustainability Boundary Approach (SBA)". Francisco et al. used SBA to assess and manage three major river basins in Spain with good results [15,16]. Typical reservoir operating rule curves (RORCs) [17] were improved by using RVA as evaluation indicators [18,19].

While it is now recognized that a comprehensive evaluation of ecological disruption requires an investigation over the entire river system, the water need in river ecosystems must also be considered on multiple scales. The minimum ecological discharge is mainly reflected on a monthly scale, but pulse flow and ecological flow velocity are reflected on smaller scales. As such, studies that are restricted to any single scale are inherently missing important ecological information. That being said, the large scales used for current research on reservoir ecological operation are too high-level to describe ecosystem demand with more than an overview, as flow pattern and flood pulse, etc., are too specific to be considered.

Given that the ecological water demand is present on multiple scales, any ecological dispatching model needs to include various scales. Steinschneider et al. [20] developed a large-scale optimization model to evaluate the contribution of reservoir management measures to ecological benefits; and Tsai et al. [21] proposed a new hybrid method to quantify river ecosystem demand based on artificial intelligence. Clearly, the importance of a large-scale optimization model and small-scale characteristic flow regimes has been recognized, but few studies couple them together in a feasible and practical way. Although the multi-scale dispatching model covers more ecological information, the number of model decision variables dramatically increases accordingly. For example, the number of decision variables in this study is nearly 3000. Increasing the number of model decision variables and the non-linear forms of hydrologic alteration makes the overall system model difficult to solve and even harder to understand. Furthermore, current studies usually employ larger step sizes (such as ten-day period) and take scheduling rules parameters as decision variables [17], but they fail to acquire all the characteristics of daily flows. Finally, the large-scale dispatching model is unable to be learned by machine-learning algorithms. In order to overcome these challenges, a hierarchical control method of the decomposition-coordination principle was applied in this study. The large system decomposition coordination theory is a new branch of science that has developed rapidly since the 1970s. In its most fundamental form, the theory proposes decomposing a complicated system into several independent subsystems and handling the correlation between subsystems by coordinators. In 1984, Adigüzel [22] decomposed a complicated system into several subsystems according to the objective function based on the decomposition-coordination principle. Currently, the decomposition-coordination principle is one of the most commonly used methods for solving complex

systems. Over time, as the decomposition-coordination principle was more frequently applied, a multi-layer hierarchical control system was established. In the multi-layer hierarchical control system, the superior subsystem result is not only the boundary but is also harmonized with the result of the subordinate subsystem.

As stated above, researchers acknowledge the multi-scale features of ecological water demand [23,24]; yet few studies have been performed on the multi-scale coupled ecological dispatching model. Thus, in order to obtain a more reasonable deterministic optimization model, a multi-scale coupled ecological dispatching model based on the decomposition-coordination principle was built in this study. The scheduling model consists of large-scale and small-scale models, which use the month and day, respectively, as the calculation scale. The large-scale model was implemented for solving problems associated with water supply and river ecological base flow; while the small-scale model was used for solving problems related to extremely low flows, flood pulses, and ecological hydraulic parameters. Due to the introduction of RVA, disturbances are kept within reasonable bounds, which is beneficial to the river ecosystems. Furthermore, the simulative daily flows can be applied to the training of ecological regulations by a machine learning algorithm, which is important to the implicit stochastic optimization. Investigations that take scheduling rules parameters as decision variables or use larger step sizes cannot obtain valid daily flow data. The multi-scale coupled model also provides a framework for the formulation of multi-scale scheduling rules. This model can help decision-makers to determine whether there is still room for improvement in ecological scheduling and look for ways to improve.

## 2. Materials and Methods

### 2.1. Generation and Evaluation of Characteristic Flow Regimes

The importance of natural flow regime to the riverine ecosystems has been widely accepted. Richter et al. [4,25] put forward IHA and RVA indicators to quantify the degree of flow regime alteration. Consequently, the effect of reservoir ecological operation can be quantified. Mathews and Richter [26] suggested that flows can be divided into five categories: small floods (flows ≥ the overbank flow, but <10 year flood), high-flow pulses (flows < the overbank flow, but > seasonal base flow), low flows (base flows in different months), extreme low flows (flows ≤ 95th percentile flow) and large floods (flows ≥ 10 year flood). Yin et al. [19] combined small floods and large floods into one flood category. Furthermore, to consider additional physical mechanisms, we take into account the hydraulic parameters that do not belong to the IHA in this study.

In order to build a comprehensive multi-scale coupled ecological dispatching model, the flow regime alteration level must be quantified, and the approaches for generating the characteristic flow regime must also be explained. In this study, all environmental flow components were accounted for by considering four kinds of characteristic flow regimes: base flows, extreme low flows, flood pulse with high flow pulses, and ecological hydraulic parameters. The generation mechanisms for these characteristic flow regimes are based on their ecological significance.

### 2.1.1. Base Flows

Ecological functions of base flows include maintaining suitable water quality, sustaining hydraulic habitats for aquatic organisms, and supporting hyporheic organisms [19]. Thus, maintaining base flow in a river is very important for sustaining the riverine ecosystem. Based on the ecological functions of base flow, the full spectrum of base flow magnitudes was selected as evaluation indicators in the multi-scale coupled ecological dispatching model.

Depending on the dates in question, a method (e.g., Tennant or 7Q10 methods, etc.) was selected and implemented to calculate the base flows. The base flows were then used as constraint conditions in the large-scale scheduling model, which has an account step of 1 month. Merging the large- and small-scale factors guaranteed that monthly discharge ≥ base flows.

With respect to the model's objective functions, RVA was used to quantify the degree of deviation between the discharged flow's base flow and the natural flow regime. In this study, the daily flow from 1958 to 1977 were used as the reference series—as the values closely reflect natural runoff; 2007–2014 was selected as the scheduling period. In order to avoid ecological flow flattening, the full spectrum of base flows within the reference period was divided into three target ranges using the 25th and 75th percentiles as parameter values. The parameters in every target range were preserved. The corresponding target ranges were integrated into the objective functions and mathematically expressed as follows:

$$D_i = \sum_{La=1,2,3} \left| \frac{N_i(La) - N_e(La)}{N_e(La)} \right| \tag{1}$$

where $La$ = 1, 2, 3 represents the high, middle and low target ranges, respectively; $D_i$ refers to the degree of change of the *i*-th index; $N_i(La)$ refers to the number of years in which the i-th index is observed in the corresponding La target ranges; $N_i(La)$ refers to the number of years in which the i-th index is expected to be observed in the corresponding $La$ target ranges which were calculated according to the reference series.

As the number of parameters from the simulation period that are observed in the target ranges increases, the degree of hydrologic alteration decreases in parallel.

### 2.1.2. Extreme Low Flows

Extreme low flows concentrate prey species for predators, provide habitat for terrestrial animals, and purge exotic species. At the same time, the extremely low flows can reduce the connectivity of rivers and confine the movement of certain aquatic organisms.

For ecological reasons, the magnitude, frequency, duration, and timing of extremely low flows were taken into consideration in the small-scale scheduling model. As with base flows, the full spectrum of the parameters (the magnitude, frequency, duration, and timing) was grouped into three target ranges using the 25th and 75th percentiles as parameter values. The mathematical expression is identical to that shown in Formula (1). Just as with the base flows, the more parameters from the simulation period that are observed in target ranges, the lower the degree of hydrologic alteration.

### 2.1.3. Flood Pulses and High Flow Pulses

Junk [27] introduced the Flood Pulse Concept (FPC). Flood pulse is the primary driving force behind biological survival, productivity, and interaction of the river-flood plain system. The flood pulse's activity has the following ecological significance: (1) It promotes energy exchange and material circulation between aquatic species and terrestrial species; (2) it improves the dynamic connectivity of river systems, which provide shelter and habitat for different species; and (3) it transmits abundant and intense information to species, which is important for spawning, hatching, growth, refuge, and other migration activities.

Flood pulse and high flow pulse generation mechanisms are similar; they differ in threshold values. Using flood pulse as an example, in order to represent its ecological purposes, the magnitude, duration, timing, and frequency are considered in the small-scale scheduling model. Thus, the flood pulse generation mechanism is as follows:

When the control section flow exceeds a certain limit, a flood pulse incident is triggered.

$$Q_T > T \tag{2}$$

where T is a pre-set limit value that depends on the stage-discharge relationship and the control section shape; and $Q_T$ is the control section flow.

When the control section flow decreases below a certain limit, the flood pulse incident ends. In the small-scale scheduling model, a flood pulse is induced by artificial floods due to the release flow from reservoirs. Thus, reservoirs need to control the release to compensate for the flood pulse as follows.

$$Q_P = Q_R - Q_T \tag{3}$$

where $Q_P$ represents the released flow from reservoirs; $Q_R$ represents the control section target flow; and $Q_T$ represents the control section flow.

Since flood pulse incidents usually last for a few days or less, conventional reservoir scheduling models with 1 month account steps, cannot incorporate the requirement. Thus, the magnitude, frequency, timing, and duration of all flood pulse incidents during the ecological critical period were evaluated in the small-scale scheduling model. The mathematical expression is the same as that shown in Formula (1).

### 2.1.4. Ecological Hydraulic Parameters

Using hydraulic methods to calculate ecological water demand is both prominent and important. Hydraulic methods are based on hydrodynamic parameters as well as specific biological criteria. Conventional hydraulic methods include the wet perimeter method and R2-CROSS [7,28–30]. Liu et al. [31] introduced the concept of ecological flow velocity, which combines hydrodynamic parameters and specific ecological targets.

Ecological flow velocity differs from the above-mentioned characteristic flow regimes in that the ecological hydraulic parameters reference standard is based on both the natural flow regime and species requirements. Because ecological flow velocity reflects the ecological target requirement, it is expected to vary within a certain range. In addition, specific water depths—which will differ depending on the local channel morphology and the local ecology—are essential for maintaining the basic ecological function of the ecosystem. In order to fully incorporate the biological requirements and hydrodynamic parameters, this study accounts for the appropriate hydraulic parameters, such as the depth and flow velocity of specific species in different growth cycles. The selected species primarily represent those that dominate the riverine ecosystem. For example, the growth cycle of *Gymnocypris eckloni herzenstein*—a keystone species found upstream of the Yellow River—was divided into three stages: spawn, brood, and growth. Liu [32] determined that maintaining a larger flow velocity during the spawning period stimulates spawning. In contrast, slow to moderate flow velocity is optimal for the brood or growth stage. In terms of depth, Xu [33] established that fish require a minimum water depth more than three times of their length to survive (See Table 1).

**Table 1.** Gymnocypris eckloni herzenstein.

|  | Length/cm | Suitable Flow Velocity/(m/s) | Critical Flow Velocity/(m/s) |
|---|---|---|---|
| *Gymnocypris* | 20–25 | 0.3–0.8 | 1.00 |
| *eckloni herzenstein* | 25–35 | 0.3–0.8 | 1.10 |

Table 1 shows ecological parameters of the *Gymnocypris eckloni herzenstein*. This data is from the Advanced Aquatic Biology [32].

Flow was related to the hydraulic parameters via an open channel uniform flow theory. The conventional reservoir scheduling models were unable to meet the requirement because the hydraulic parameters are sensitive to flow rate. Therefore, the small-scale scheduling model uses a day scale as the account step. The ecological hydraulic parameters evaluation function is the same as that shown in Formula (1). However, in this case, the target range is based on the natural flow and species requirements. The more hydraulic parameters, from within the corresponding natural flow values that are observed in the suitable ranges, the lower the degree of hydrologic alteration that will be obtained.

The descriptive parameters of the four discussed characteristic flow regimes are summarized in Table 2:

**Table 2.** Characteristic flow regimes.

| Characteristic Flow Regimes | Descriptive Parameters | Hierarchy of Model |
|---|---|---|
| Base Flows | Magnitude of base flows | Large-scale |
| Extreme Low Flows | Magnitude, frequency, duration, and timing of extremely low flows | Small-scale |
| Flood Pulses and High Flow Pulses | The magnitude, frequency, duration, and timing of flood and high flow pulses | Small-scale |
| Ecological Hydraulic Parameters | The hydraulic parameters for specific species in different growth cycles | Small-scale |

Table 2 shows descriptive parameters of characteristic flow regimes and which model will handle them.

In this study, the characteristic flow regimes' degree of alteration was averaged and used as the degree of hydrologic alteration.

### 2.2. Construction of the Multi-Scale Coupled Ecological Dispatching Model

Two key problems with the multi-scale coupled model include: (1) increased decision variables and (2) coordination between the different scale models. In order to resolve these issues, the decomposition-coordination principle and multiple hierarchical control system, which is based on the theory of large-scale systems, were applied to this work.

### 2.2.1. The Space and Time Decomposition-Coordination Method

To resolve the issue of increased decision variables due to the introduction of small-scale scheduling models, the whole basin scheduling model for the entire scheduling period was divided into numerous subsystems, each of which includes part basin and part scheduling period. Because the whole basin scheduling model is regarded as a large complex system, the connection between the overall system and subsystems is reflected by the subsystems' objective functions, which include Lagrange multipliers. In this way, the large system optimization problem is transformed into several optimization problems of the subsystem objective functions. This method greatly reduces decision variables while retaining the model's optimal conditions [34]. With respect to space, the large-scale system is usually divided by reservoir control basins. In terms of time, the entire scheduling period is divided into small periods according to the model requirements. In general, subsystems coordinate with each other via subsystem hydraulic connections and cooperative work between subsystems—i.e., the interconnection constraints. The subsystem is mathematically expressed as follows:

$$L = \sum_{i=1}^{n} L_i = \sum_{i=1}^{n} f_i(x_i) + \lambda \left[ \sum_{i=1}^{n} g_i(x_i) - b \right] + \mu \left[ \sum_{i=1}^{n} h_i(x_i) - d \right]$$
$$\sum_{i=1}^{n} g_i(x_i) = b \qquad (4)$$
$$\sum_{i=1}^{n} h_i(x_i) \geq d$$

where $L$ represents the large system Lagrange function; $L_i$ represents the Lagrange function of the i-th subsystem; n represents the number of subsystems; $x_i$ represents the decision vector of the i-th subsystem (in this study, it represents water released from reservoirs); $f_i(x_i)$ represents the evaluation function of the i-th subsystem; $\sum_{i=1}^{n} g_i(x_i) = b$ is the equality constraint between subsystems; $\sum_{i=1}^{n} h_i(x_i) \geq d$ is the inequality constraint between subsystems; $\lambda$ is the equality Lagrange multiplier constraint; and $\mu$ is the inequality Kuhn-Tacker multiplier constraint.

As shown in Equation (4), the equality constraint and inequality constraint are frequently used for constraining water balance and constraining cooperative work, respectively.

In this work, the subsystems' optimal solutions were solved using an adaptive genetic algorithm (AGA). The coordination between subsystems was implemented by means of coordinating variable iterations (Lagrange multiplier or Kuhn-Tacker multiplier) within the coordination level. The mixing method, which includes the interaction balance method and interaction prediction method, was employed for solving coordinating variables.

### 2.2.2. Coordination between Multi-Scale Subsystems

It is clear that the subsystems' hydraulic connections and cooperative work between subsystems are often regarded as interconnection constraints between subsystems of the same scale. The multi-scale coupled ecological dispatching model accounts for the coordination between different scale subsystems. At first, the water balance constraint is the only interconnection constraint in the multiple hierarchical control system:

$$\sum_{t=t_0}^{t_f} x_s(t) = x_i \tag{5}$$

where $t_0$ and $t_f$ are the start and finish time of subordinate subsystems, respectively (smaller scale); $x_s$ represents the subordinate subsystem decision variable; and $x_i$ is the result of superior subsystem (larger scale).

As shown in Equation (5), the sum of the discharge flows in the small-scale model is equal to the discharge flow in the large-scale model for the corresponding period. Clearly, the water balance constraint is an equality constraint. Therefore, a new Lagrange multiplier, $\lambda_s$, was introduced to regulate the water balance constraint. Usually, the Lagrange equation L is used as the whole basin scheduling model objective function. Thus, the mathematical expression of the multiple hierarchical control system is as follows:

$$L = \sum_{i=1}^{n} \left\{ f_i(x_i) + \sum_{t=t_0}^{t_f} f_s(x_s) + \lambda_s \left[ \sum_{t=t_0}^{t_f} x_s(t) - x_i \right] \right\} + \lambda \left[ \sum_{i=1}^{n} g_i(x_i) - b \right] + \mu \left[ \sum_{i=1}^{n} h_i(x_i) - d \right]$$
$$\sum_{i=1}^{n} g_i(x_i) = b$$
$$\sum_{i=1}^{n} h_i(x_i) \geq d \tag{6}$$
$$\sum_{t=t_0}^{t_f} x_s(t) = x_i$$

where $\lambda_s$ is the Lagrange multiplier that regulates the water balance constraint between subsystems of different scales; $f_s(x_s)$ represents the subordinate subsystem's (smaller scale) evaluation function belonging to the i-th subsystem. For our purposes, b represents the vector of the initial water amount in the reservoir at each time; d represents the vector of common water supply mission between reservoirs. The other symbols are the same as those defined in Equations (4) and (5).

As demonstrated in the above formulas, subsystems of different scales have different evaluation functions. However, solving a hierarchical control model with different objective functions for different layers is very difficult. As such, in this work, large-scale subsystem objective functions were converted to the subordinate subsystem's constraint conditions. In this way, objective functions of the large-scale subsystems were implemented by the subordinate subsystems. Unlike common multi-scale models that use results of the superior subsystem merely for subordinate subsystem boundary conditions, multi-scale subsystems—which are based on the theory of large-scale systems—consider coordination between different scale subsystems via an iteration of the coordination variables (Lagrange multiplier or Kuhn-Tacker multiplier).

The implication of coordination variables is deduced by the following interaction prediction method:

$$Let: \frac{\partial L}{\partial \lambda_s} = \sum_{t=t_0}^{t_f} x_s(t) - x_i = 0 \tag{7}$$

In order to satisfy Karush–Kuhn–Tucker conditions [35]:

$$\frac{\partial L}{\partial x_i} = \frac{\partial f_i}{\partial x_i} - \lambda_s + \lambda \frac{\partial g_i}{\partial x_i} + \mu \frac{\partial h_i}{\partial x_i} = 0 \tag{8}$$

$$\frac{\partial L}{\partial [\sum_{t=t_0}^{t_f} x_s(t)]} = \frac{\partial [\sum_{t=t_0}^{t_f} f_s(x_s)]}{\partial [\sum_{t=t_0}^{t_f} x_s(t)]} + \lambda_s = 0 \tag{9}$$

Thus:

$$\lambda_s = \frac{\partial f_i}{\partial x_i} + \lambda \frac{\partial g_i}{\partial x_i} + \mu \frac{\partial h_i}{\partial x_i} = -\frac{\partial [\sum_{t=t_0}^{t_f} f_s(x_s)]}{\partial x_i} \tag{10}$$

As shown in the above formula, the large system global optimal solution is obtained exclusively at the moment when the derivative of the subordinate subsystem to the decision variable and the derivative of the superior subsystem to the decision variable are equal.

### 2.2.3. Construction of the Subsystems

After a coordination level was assigned for the coordination variables, the subsystem was converted into a relatively simple problem. Hence, AGA was used to mathematically solve the subsystem. The subsystem's objective function is written as follows:

$$L_i = f_i(x_i) - \alpha \lambda_s x_i + \lambda_i g_i(x_i) + \mu_i h_i(x_i) \tag{11}$$

where $L_i$ represents the Lagrange function of the i-th subsystem; $f_i(x_i)$ represents the evaluation function of the i-th subsystem; $\alpha$ is a 0-1 variable, which is used as a subordinate subsystem structural discrimination coefficient, 1 and 0 denote whether the i-th subsystem has a subordinate subsystem or not, respectively; $\lambda_i$ and $\mu_i$ refer to the corresponding coordination variable components of the i-th subsystem; and the other symbols are the same as those defined in Equations (4) and (5).

The mathematical expression of $f_i(x_i)$ is as follows:

$$f_i(x_i) = (1-\alpha)\left(\sum_{t=1}^{T} \frac{W - x_i}{W} + \sum_{j=1}^{M} D_j\right) + \alpha \sum_{j=1}^{N} D_j \tag{12}$$

where W refers to the subsystem water demand; $x_i$ refers to the subsystem water supply; $D_j$ refers to the degree of change of the j-th index; M and N represent the number of superior subsystem and subordinate subsystem indicators, respectively; and the other symbols are the same as those defined in Equations (4) and (5).

### 2.3. A Solution of the Multi-Scale Coupled Ecological Dispatching Model

The gradient method was used to mathematically resolve the coordination level. The coordination variables were all initialized to zero; and the subsystem's calculation results were passed to the coordination level. The iterative formulas are as follows:

$$\sum_{i=1}^{n} g_i(x_i) = b \tag{13}$$

$$\sum_{t=t_0}^{t_f} x_s(t) = x_i \tag{14}$$

$$\lambda_s^{j+1} = -\frac{\partial[\sum_{t=t_0}^{t_f} f_s(x_s)]}{\partial x_i} \tag{15}$$

$$\mu_i^{j+1} = \mu_i + [h_i(x_i) - d] * R \tag{16}$$

$$\lambda_i^{j+1} = \frac{\partial f_{i+1}}{\partial x_i} - \lambda_{s_{i+1}} + \lambda_{i+1} + \mu_{i+1} \tag{17}$$

where the symbol superscripts represent the number of iterations; unmarked symbols represent symbols of the j-th iteration; symbol subscripts represent the number of subsystems; R refers to the account step of the coordination variables; $\frac{\partial f_{i+1}}{\partial x_i}$ represents the derivative of the objective function of the i + 1-th subsystem to initial water of this subsystem (equal to surplus water of the last subsystem); $\frac{\partial[\sum_{t=t_0}^{t_f} f_s(x_s)]}{\partial x_i}$ represents the derivative of the objective function of the inferior subsystem to the superior subsystem output; and the other symbols are the same as those defined in Equations (4)and (5).

After the calculation, the coordination variables were passed to the subsystems until the convergence condition was met. The coordination variables' convergence conditions are expressed as:

$$\left|\lambda_s^{j+1} - \lambda_s^{j}\right| \le \varepsilon_{\lambda_s} \tag{18}$$

$$\left|\mu_i^{j+1} - \mu_i^{j}\right| \le \varepsilon_{\mu_i} \tag{19}$$

$$\left|\lambda_i^{j+1} - \lambda_i^{j}\right| \le \varepsilon_{\lambda_i} \tag{20}$$

where $\varepsilon_{\lambda_s}, \varepsilon_{\mu_i}, \varepsilon_{\lambda_i}$ refer to the coordination variables' convergence thresholds.

Unlike the traditional targets, the derivative of the ecological objective function to initial water is difficult to calculate. Thus, in this work, a differential response model between the ecological objective function and initial water of a small-scale subsystem was established. The results were calculated in advance for coordination level.

### 2.4. Description of the Study Area

The lower reaches of the Yellow River (Figure 1) refer to the Yellow River stream segment below Peach Blossom Valley. At this location, the river length is 786 km and the drainage area covers 23,000 km$^2$. Furthermore, the stream gradient is small and the river's ecological problems are serious. The main water conservancy projects within this area are the Xiaolangdi reservoir and the Sanmenxia reservoir; which effectively store 5.1 billion cubic meters (bcm) and 0.462 bcm of water, respectively. Annual consumptive water use in the study area is 7.5 bcm. During flood season, there are water resources to ensure the implementation of ecological operation.

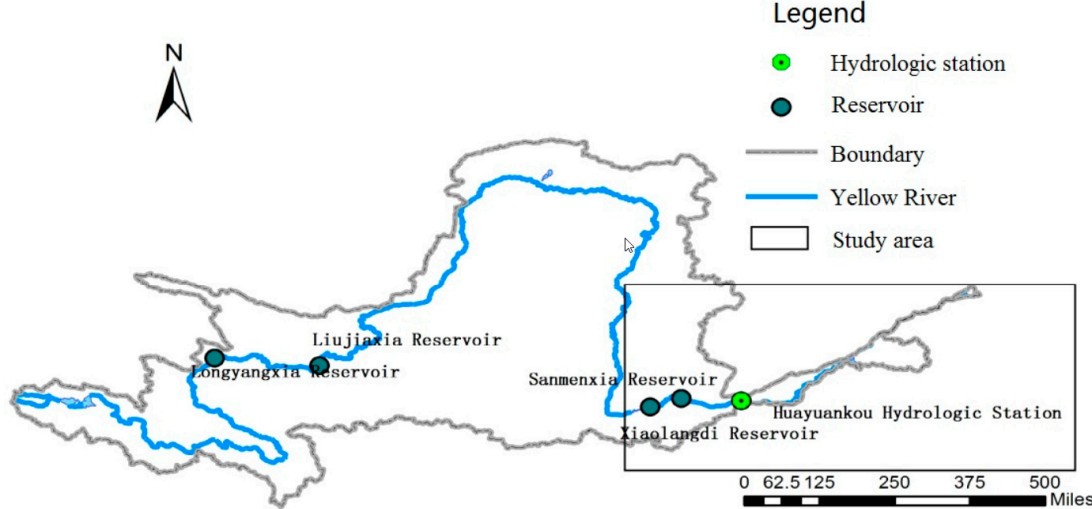

**Figure 1.** Locations of the downstream region of the Yellow River.

The ecological problems in the lower reaches of the Yellow River differ depending on the time of year. (1) During the major flood period—from July to September—the main ecological problem is the disappearance of the flood and high flow pulses, resulting in a flat, non-fluctuating flow; (2) during the drought period—from November to March—the river suffers a water shortage and fails to meet the minimum instreaming ecological flow requirements; (3) during medium season, the primary problem is habitat change in the riverine biological communities.

Because the characteristic flow regimes are relatively complete in the major flood period, for this study, the major flood period was selected as the ecological critical period that was assessed using the small-scale scheduling model. Water demand data used in this work came from the Comprehensive Planning of the Yellow River Basin [36], which is a book compiled by the Yellow River Conservancy Commission (YRCC) of the ministry of water resources of China; and the daily runoff data was taken from the Hydrological Yearbook of the People's Republic of China [37].

The multi-scale coupled ecological dispatching model shown in Figure 2 was established based on the characteristics of the study area described above.

In Figure 2, x refers to the discharge from the reservoirs; $\lambda_1$ and $\lambda_2$ are the Lagrange multipliers that regulate the water balance constraint between the Xiaolangdi and Sanmenxia reservoir superior subsystems, respectively, at different times; $\lambda_3$ and $\lambda_4$ are the Lagrange multipliers that regulate the water balance constraint between the superior subsystems and subordinate subsystems, respectively, of the Xiaolangdi and Sanmenxia reservoirs; and $\mu$ is the Lagrange multiplier that regulates the water balance constraint between the superior subsystems during the same period.

The objective function of the model is as follows:

(1)  Minimum water shortage rate:

$$Minf_{S-D} = \sum_{t=1}^{T} \sum_{m=1}^{M} \frac{D_{(m,t)} - S_{D(m,t)}}{D_{(m,t)}} \tag{21}$$

where $f_{S-D}$ refers to the objective function of the water deficient ratio; m refers to the serial number of the reservoir; M refers to the sum of reservoirs; t refers to the scheduling period number; T refers to scheduling period length; $D_{(m,t)}$ refers to water demand of reservoir m at the t-th period; and $S_{D(m,t)}$ refers to the water supply of $D_{(m,t)}$;

(2)　The minimum degree of hydrologic alteration is expressed as:

$$Min\sum_{i=1}^{I} D_i = \sum_{L=1,2,3} \left| \frac{N_i(La) - N_e(La)}{N_e(La)} \right| \tag{22}$$

where I represents the total number of indicators; La = 1, 2, 3 represents the high, middle and low target ranges, respectively; $D_i$ refers to the degree of change of the i-th index; $N_i(La)$ refers to the number of years in which the i-th index is observed in the corresponding target ranges L; $N_e(La)$ refers to the number of years in which the i-th index is expected to be observed in the corresponding target ranges L.

The objective function (Lagrange function) of the overall system model is obtained by substituting Formula (21) and Formula (22) into Formula (6).

Constraint conditions of the model include water level constraint, flow constraint, water balance constraint, and flow balance constraint.

(1)　Water level constraint:

$$Z_{\min}(m,t) \leq Z(m,t) \leq Z_{\max}(m,t) \tag{23}$$

(2)　Flow constraint:

$$Q_{O\min}(m,t) \leq Q_O(m,t) \leq Q_{O\max}(m,t) \tag{24}$$

(3)　Water balance constraint:

$$V(m,t+1) = V(m,t) + Q_I(m,t) - Q_O(m,t) \tag{25}$$

(4)　Flow balance constraint:

$$Q_I(m+1,t) = Q_O(m,t) + q(m,t) \tag{26}$$

(5)　Water supply reliability constraint:

$$reliability_{agricultural} \geq 0.75$$
$$reliability_{urban} \geq 0.95 \tag{27}$$

where $Z(m,t)$ refers to the water level of reservoir m at period t; $Z_{\max}(m,t), Z_{\min}(m,t)$ are the top and bottom limitations of the water level of reservoir m, respectively at period t; $Q_O(m,t)$ refers to the discharged flow of reservoir m at period t; $Q_{O\max}(m,t), Q_{O\min}(m,t)$ are the top and bottom limitations of the discharged flow of reservoir m at period t, respectively; $V(m,t)$ refers to the storage of reservoir m at period t; $Q_I(m,t)$ refers to the inflow of reservoir m at period t; $q(m,t)$ refers to the local inflow of reservoir m at period t; and $reliability_{agricultural}$ and $reliability_{urban}$ refer to the reliability of agricultural and urban water supply, respectively (urban water supply includes the industrial water supply and domestic water supply). The reliability data comes from the Comprehensive Planning of the Yellow River Basin.

Among the constraint conditions, water balance constraint between scheduling periods and flow balance constraint were implemented by coordination variables.

In this study, the daily flow from 1958 to 1977 was used as the reference series and the daily flow from 2007 to 2014 was selected as the scheduling period. The reservoir inflow, domestic water demand, agricultural water demand, and ecological water demand, etc., are required inputs of the multi-scale coupled ecological dispatching model.

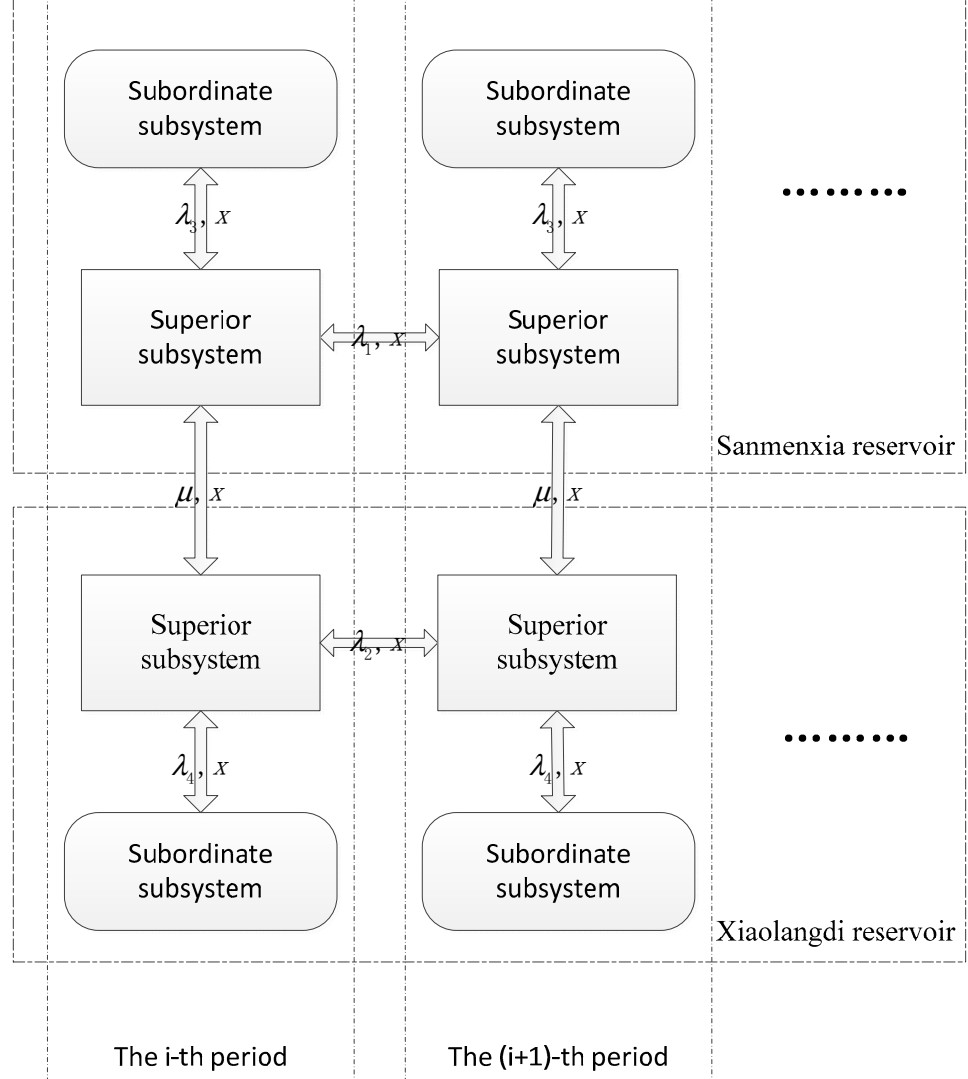

**Figure 2.** Information transfer structure diagram of the model.

## 3. Results

For comparison purposes, the hydrologic alterations caused by the operation of the current reservoir in the middle and lower Yellow River were assessed using the RVA method. The daily flow from 1958 to 1977 was used as the pre-impact series, and 2007–2014 were selected as the post-impact series. The degree to which the RVA target range is not attained is a measure of hydrologic alteration and is calculated as follows:

$$(|\text{Observed frequency} - \text{Expected frequency}|)/\text{Expected frequency} \tag{28}$$

The RVA uses 33 hydrological parameters to evaluate the hydrologic alterations [4], which are categorized into five groups: magnitude of monthly water conditions, magnitude and duration of annual extreme conditions, timing of annual extreme water conditions, frequency and duration of high and low pulses, and rate and frequency of water condition changes. As shown in Figure 3, the integrated hydrologic alteration reaches 86%. In terms of indicators, most change was observed in the magnitude of monthly water conditions and the magnitude and duration of annual extreme conditions. Among those, the change in magnitude of monthly water conditions during the flood season is the largest. In terms of RVA categories, the high RVA categories reduced the most.

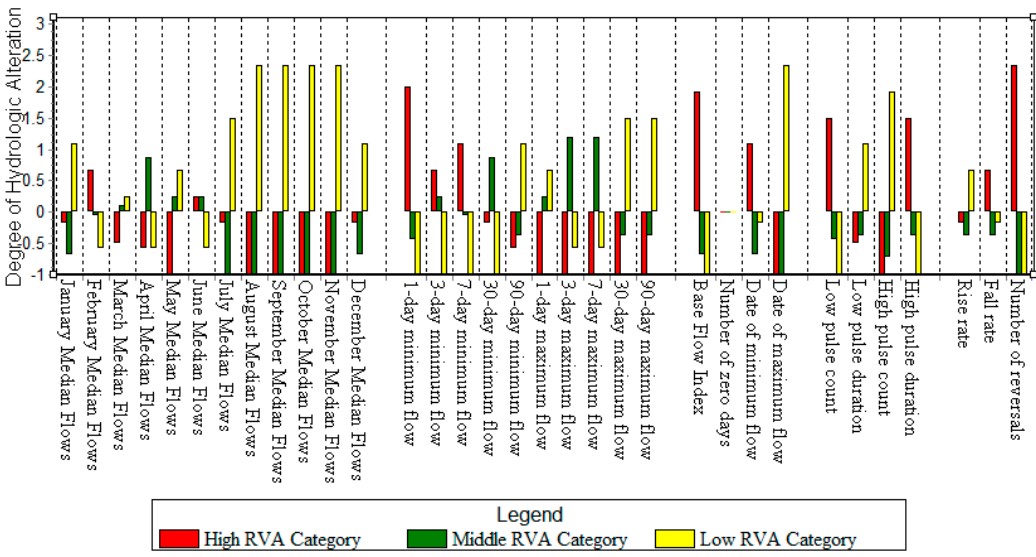

**Figure 3.** Hydrologic alteration of current scheduling.

The multi-scale coupled ecological dispatching model was established to generate small-scale ecological flow regimes and avoid dimension disaster. In this study, subsystems were solved by AGA and coordinated via the coordination variables. As such, the degree of hydrologic alteration can be used as an evaluation function of the reservoir scheduling model, which is unattainable for the conventional models. Thus, the multi-scale coupled ecological dispatching model provides more information to reservoir managers. Compliments of decomposition-coordination algorithms and AGA, large-scale model and small-scale model results can efficiently converge to the optimal solutions within 200 iterations as shown in Figure 4.

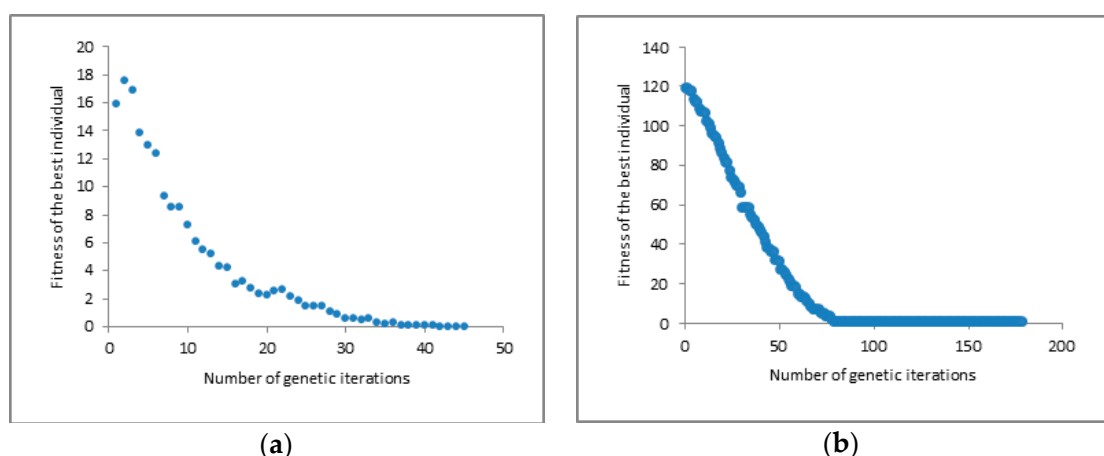

**Figure 4.** (**a**) Large-scale model optimization processes; (**b**) Small-scale model optimization processes.

Figure 5 compares the simulative monthly flow and actual monthly flow from 2007 to 2014. The results show that the model is capable of solving the challenges associated with uneven spatial and temporal distribution of water resources. Three observations are evident in Figure 6: (1) The simulative monthly flow is more fluctuant than the actual monthly flow; (2) the simulative monthly flow in the flood season is larger than the actual monthly flow; and (3) the occurrence time of the simulative high flow is more regular than actual high flow. These results are explicable, as more water was needed during the ecologic critical period to generate the characteristic flow regimes. Moreover, the average flow in flood season can be increased to 1400 m³/s under the premise of meeting the water supply reliability.

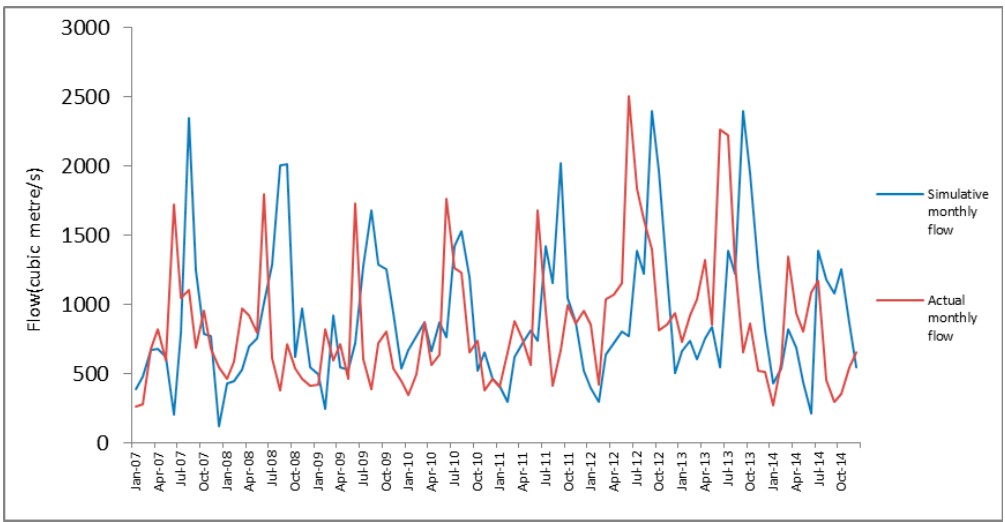

**Figure 5.** Monthly discharge from Xiaolangdi reservoir.

The 'percent-of-flow' (POF) approach has been widely used in the USA, the European Union, and elsewhere over the past decade [3]. To contrast the above results, the flow was also generated using the POF approach, which sets protection standards by using allowable departures from natural conditions, expressed as percentage alteration. The hydrologic alteration of flow was calculated under the POF approach (Figure 6) and contrasted with the hydrologic alteration of simulative flow (Figure 6). Comparison of Figures 3 and 6 shows that the POF approach is more protective with respect to the magnitude of monthly water conditions—as the Hydrologic alteration of the magnitude of monthly water conditions were reduced from 81% to 65%. Yet, in other groups, particularly in the magnitude and duration of annual extreme conditions, the POF approach does not provide a high degree of protection for natural flow variability. Contrarily, the hydrologic alteration of the simulative flow was reduced in all five groups. The integrated hydrologic alteration was reduced from 86% to 53%. On the one hand, this is because the objective function of the optimization model is to reduce the degree of hydrological alterations. On the other hand, the incoming water in the whole dispatching period of the optimization model is known, and this is not possible in the actual scheduling process. The hydrologic alterations of five groups are 0.60, 0.41, 0.29, 0.67, and 0.67, respectively (see the beginning of this section for corresponding groups). It is worth noting that the simulative flow did not provide a very high degree of protection for the frequency and duration of high pulses, particularly in the high RVA categories. This is explained by the difficulties incurred with providing protection for natural flow variability of different indicators in different RVA categories. For instance, more water was needed to keep the frequency of flood pulses in the high RVA category; and the increased water concentration reduced the unit water value, resulting in a lack of water elsewhere.

From the perspective of hydraulic parameters, the number of days within the appropriate velocity range during the reference series (1958–1977) is 2117, which is 29% of the total number of days. The number of days within the appropriate velocity range during the scheduling period (2007–2014) is 1452, which is 50% of the total number of days. Due to the influence of flow flattening, the number of days suitable for fish has increased. We expected that the duration of the simulated flow within the appropriate velocity is not less than the duration of natural flow. The number of days within the appropriate velocity range in the simulated flow accounted for 46% of the total. And the number of days within the appropriate velocity range under the POF approach accounted for 43% of the total. Obviously, this is an easy goal to achieve.

To solve the problem of uncertainty, we use the autoregressive model to generate nine sets of inflow runoff series. Ten sets of inflow runoff series (Including the actual inflow series from 2007–2014) were used as input for simulation calculation. The final result is the average of 10 sets of calculated results and the variation range of 10 sets of results. The values above are the mean, and the values in

brackets are the variation ranges. The quantitative evaluation and biological significance of various indicators under the POF approach and optimization model respectively are shown in Table 3. As it is difficult to simulate decision-makers' decisions, the uncertainty of current scheduling is not shown. The hydrological alterations of the current scheduling are only used as a reference in Table 3. We found that the calculation results of the optimization model performed well in the second, third and fifth categories of indicators. That means the model performed better in terms of vegetation expansion, fluvial topography, natural habitat construction, fish migration, and spawning and life cycle reproduction compared to current scheduling and the POF approach. In terms of uncertainty, the optimization model performs better than the POF approach too. Especially in the second, third and fifth categories of indicators, the performance of the optimization model are far better than the POF approach, and the variation ranges barely overlap. That means the model performed better in addressing uncertain problems compared to current scheduling and the POF approach in terms of the above ecological problems.

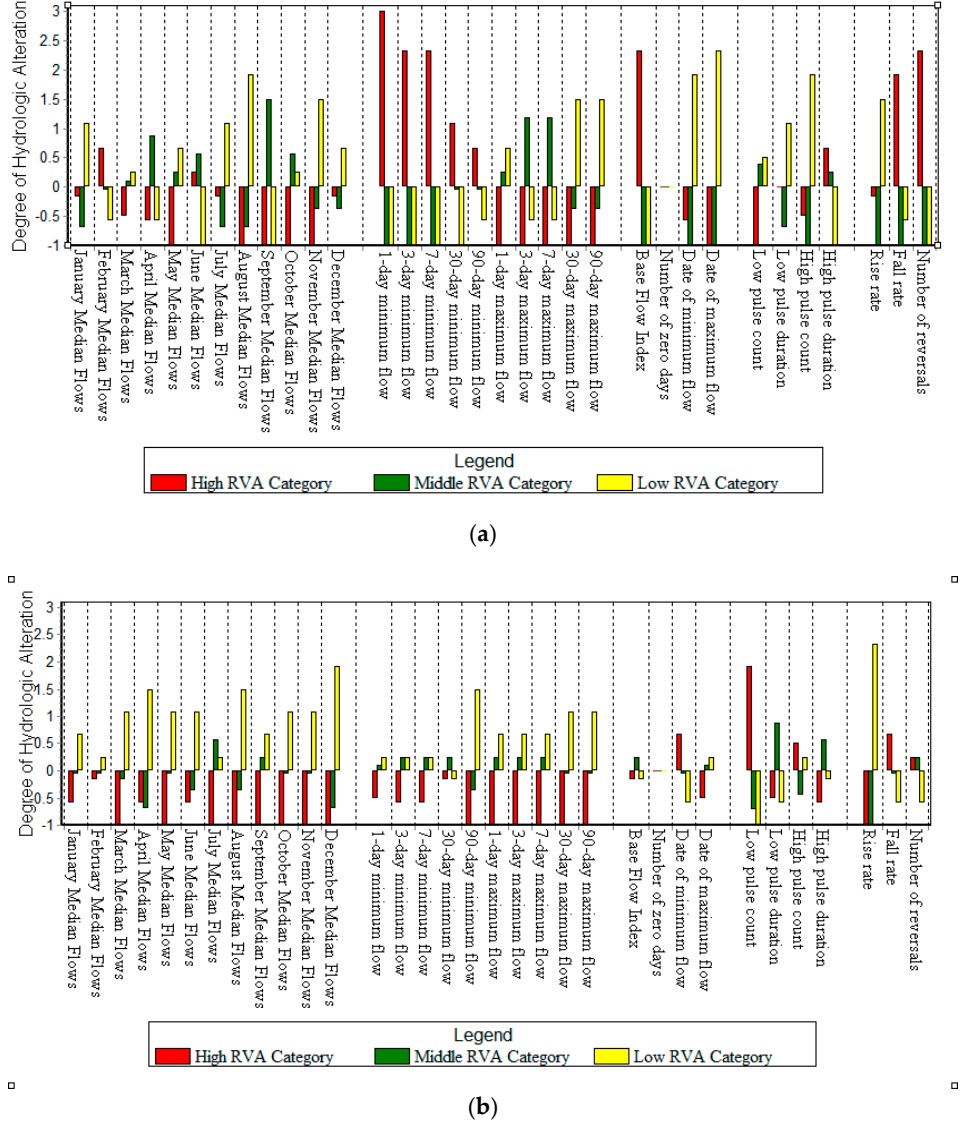

**Figure 6.** (**a**) Hydrologic alteration of POF approach; (**b**) Hydrologic alteration of simulative flow.

**Table 3.** Quantitative evaluation of the results.

| Index | Current Scheduling | POF Approach | Optimization Model | Ecosystem Influences [38] |
|---|---|---|---|---|
| Magnitude of monthly discharge conditions | 0.81 | 0.67 (0.62–0.71) | 0.60 (0.55–0.63) | To meet the habitat needs of aquatic organisms, the needs of plants for soil moisture content, the water needs of terrestrial organisms with high reliability, the migration needs of carnivores, and the influences of water temperature and oxygen content. |
| Magnitude and duration of annual extreme discharge conditions | 0.72 | 0.88 (0.80–0.95) | 0.41 (0.40–0.42) | To meet the needs of vegetation expansion, river topography and natural habitat construction, nutrient exchange in rivers and flood detention areas, distribution of plant communities in lakes, and ponds and flood detention areas. |
| Timing of annual extreme discharge conditions | 0.99 | 1.22 (0.88–1.44) | 0.27 (0.25–0.29) | To meet the migration of fish spawning, the cycle of life reproduction, biological breeding habitat conditions, and species evolution needs. |
| Frequency and duration of high/low flow pulses | 0.83 | 0.71 (0.68–0.7) | 0.70 (0.69–0.72) | To generate the frequency and magnitude of soil moisture stress for plants. |
| Rate/frequency of hydrograph changes | 0.71 | 1.12 (0.99–1.20) | 0.67 (0.65–0.69) | To meet the drought of plants, the trapping of organics on the island and in the flood detention area, and the drying stress of low-speed organisms. |
| Suitable ecological velocity | 0.29 | 0.40 (0.38–0.42) | 0.46 (0.43–0.49) | To meet the appropriate velocity requirements of fish. |

Table 3 shows the quantitative evaluation and biological significance of various indicators under the POF approach and optimization model. The values of the first five categories of indicators are degrees of hydrologic alterations. The value of the last indicator represents the proportion of the number of days in the suitable ecological velocity range to the total number of days.

## 4. Discussion

Compared to conventional models, the multi-scale coupled ecological dispatching model can obtain the degree of hydrologic alteration of small-scale ecological flow regimes and daily stream flow. At the same time, the results can be manipulated by adjusting the weights afforded to the degree of hydrologic alteration and other targets. In contrast, degrees of hydrologic alteration based on whether or not to consider the characteristic flow requirements and whether or not to introduce the small-scale model are shown in Table 4. When considering the characteristic flow requirements, reservoirs are required to release more water in the ecological critical period. Under these circumstances, the appropriate characteristic flows can be generated by the small-scale model. When the characteristic flow requirements are not taken into consideration, the agricultural water supply dependability increases from 75% to 90%.

**Table 4.** Contrast of the degree of hydrologic alteration.

| Whether Considering Characteristic Flow Requirements | Whether Introducing Small-Scale Model | Hydrologic Alteration | Agricultural Water Supply Reliability | Urban Water Supply Reliability |
|---|---|---|---|---|
| Yes | Yes | 53% | More than 75% | More than 95% |
| Yes | No | ≥53% | | |
| No | Yes | 87% | More than 90% | |
| No | No | ≥87% | | |

Table 4 shows the influence of considering characteristic flow requirements on guaranteed water supply reliability and hydrologic alteration.

Compared with the existing research, the existing model either directly optimizes the scheduling rules [17] or evaluates the existing daily flow process [24]. However, the rule optimization model cannot describe the small scale runoff process in detail, and the runoff evaluation model cannot directly guide reservoir operation. Therefore, a modelling approach which allows scheduling of reservoir operation in such a way that hydrological alteration is reduced was presented in this work. The multi-scale coupled ecological dispatching model provides a framework for the formulation of multi-scale scheduling rules. The more flexible decoupling method can be used by the model to optimize the scheduling rules directly. In further research, the large-scale model can optimize the operation rules, while the small-scale model can optimize the discharge process of reservoirs. This method is suitable for areas where the hydrological regime is seriously affected by dam operation. This model is more suitable for large rivers because of its attention to the degree of hydrological alterations. For other river basins, decision makers can choose the key ecological issues of the river basin, and the weight of the objective function can be formulated according to the key ecological problems and the ecological significance of various indicators. Scheduling rules suitable for this basin can be found by this way.

However, the multi-scale coupled ecological dispatching model established in this study still has some issues. The competitive relationship between the ecological and economical target was coordinated by the weighting method. A more reasonable and flexible multi-objective coordination mechanism should be introduced to the model. Furthermore, considering the impact of hydrological uncertainty, calculation results from the model cannot directly guide the reservoir operation. A multi-scale coupled scheduling rule for the reservoir should be put forward and planned for future studies, which is what we are going to do next.

In general, the multi-scale coupled ecological dispatching model is capable of considering the degree of hydrologic alteration of small-scale ecological flow regimes. The ecological target was regarded as both a water requirement and a difference between the artificial flow and natural flow. Operation rules extracted from results of the coupled ecological dispatching model will include more information and be more reasonable.

## 5. Conclusions

A multi-scale coupled ecological dispatching model, that considers the degree of hydrologic alteration, was established to satisfy the multi-scale features of ecological water demand. In order to acquire all characteristics of the environmental flows and to avoid the "dimension disaster", the decomposition coordination algorithm was applied to this model. Furthermore, this model is capable of using account steps from months to hours, as needed. Moreover, the general multi-scale model uses results of the superior subsystems only as boundary conditions of the subordinate subsystems; while in our model, multi-scale subsystems are based on the theory of large-scale systems and we consider the coordination of multi-scale objectives.

Simulation results show that the model is able to solve the problem of uneven spatial and temporal distribution of water resources and obtain degrees of hydrologic alteration of characteristic flow regimes. The difficulty of generating characteristic flow regimes is indicated by the results, which will be helpful to the formulation of operation rules. The calculation results indicate that increasing the average flow in flood season to 1400 m$^3$/s can reduce the degree of hydrologic alteration from 86% to 53% in the downstream Yellow River. Model optimization results were compared with results of the POF approach. The comparison showed that further reduction in hydrologic alteration is possible, but further research is needed to determine which method is more suitable for accomplishing this task.

While the proposed model is overall a success, there are some limitations worth noting. On the model side, a more reasonable and flexible multi-objective coordination mechanism needs to be established. From the mathematics side, a high-efficiency and applicative arithmetic method needs to be developed or improved to extract scheduling rules from artificial daily flow series.

**Author Contributions:** Conceptualization, T.Z. and Z.D.; methodology, T.Z.; software, T.Z.; investigation, X.G. and Z.H.; resources, X.G. and R.S.; data curation, R.S.; writing original draft preparation, T.Z.; writing—review and editing, T.Z.; W.W. and Z.D.

**Funding:** This research has been financially supported by the National Key Research and Development Program of China (2018YFC1508200), the Postgraduate Research & Practice Innovation Program of Jiangsu Province (KYCX18—0574) and the Fundamental Research Funds for the Central Universities (2018B603X14).

**Conflicts of Interest:** The authors declare no conflict of interest. The funders had no role in the design of the study; in the collection, analyses, or interpretation of data; in the writing of the manuscript, or in the decision to publish the results.

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
