# Peer review of "Study on Multi-Scale Coupled Ecological Dispatching Model Based on the Decomposition-Coordination Principle"

_water, doi:10.3390/w11071443_

Round 1

Reviewer 1 Report

The manuscript presented an important and interesting research work aiming on a very difficult task. A multiple objective optimization with multiple variables is definitely tough for any researcher.

And it might because of that, the manuscript looks too tight: too much contents in a single paper. The language problem make it even harder to follow the draft. However, the work presented here is important and exciting, the methods are well described, the results are presented clearly. For all of those, I would recommend it for publication, after a serious English editing and some presentation adjustment. There are some specific recommendations:

1)      Can the results of the case study be improved to include a operating policy that the reservoirs can follow? That will definitely be a plus for the manuscript

2)      There should be some explanation about difference between the ideal optimization and the real “every day” optimization of reservoirs, and why such difference exits. It would be helpful for readers to understand the presented work better.

3)      Lines 496-498 looks quite out of places. Are these comments supposed to be deleted but got forgotten?

Author Response

Thank you very much for your advice.

Point 1: Can the results of the case study be improved to include a operating policy that the reservoirs can follow? That will definitely be a plus for the manuscript

Response 1: It is a complicated work to make the multi-scale coupled scheduling rule. In fact we have already started this part of work and achieved some preliminary results. We use more flexible decoupling method to optimize the scheduling rules directly. But in this paper we have only made some simple scheduling recommendations that the degree of hydrologic alteration can be reduced from 86% to 53% under the condition of meeting the water supply guarantee rate in the downstream Yellow River by increasing the discharge to 1400m3/s.

Point 2: There should be some explanation about difference between the ideal optimization and the real “every day” optimization of reservoirs, and why such difference exits. It would be helpful for readers to understand the presented work better.

Response 2: On the one hand, this is because the objective function of the optimization model is to reduce the degree of hydrological alterations. On the other hand, the incoming water in the whole dispatching period of the optimization model is known. And this is not possible in the actual scheduling process. I have added this part to the article. Thank you for your suggestion.

Point 3: Lines 496-498 looks quite out of places. Are these comments supposed to be deleted but got forgotten?

Response 2: Sorry for the mistake, we have deleted this part of content.

Reviewer 2 Report

1. L349:“2.3. Description of the study area” change to “2.4. Description of the study

area”.

2. L355-356:“During flood season, there are water resources to…” change to “During

flood season, they are water resources to…”.

3. There are two spelling mistakes in Figure 1, and I suggest to use the same font in

the legend. 

4. The format of the graphic title should be consistent, “Figure 1” or “Figure 1: ”;

the font size of the graphic title should also be consistent.

5. L408:The format should be the same as before.

6. Inthe results part, I’d like to see the defined RVA target range. And I am

wondering if the hydrologic alteration values were calculated by equation 28

instead of (|Observed frequency - Expected frequency|/Expected frequency), why

a smaller sum of these values represent a smaller hydrologic alteration.

Author Response

1. L349:“2.3. Description of the study area” change to “2.4. Description of the study area”.

Response 1: Sorry for the mistake, we have modified this part of content.

2. L355-356:“During flood season, there are water resources to…” change to “During flood season, they are water resources to…”.

Response 2: We have modified this part of content.

3. There are two spelling mistakes in Figure 1, and I suggest to use the same font in the legend.

Response 3: Sorry for the mistake, we have modified the Figure 1.

4. The format of the graphic title should be consistent, “Figure 1” or “Figure 1: ”; the font size of the graphic title should also be consistent.

Response 4: We have modified this part of content.

5. L408:The format should be the same as before.

Response 5: We have modified this part of content.

6. In the results part, I’d like to see the defined RVA target range. And I am wondering if the hydrologic alteration values were calculated by equation 28 instead of (|Observed frequency - Expected frequency|/Expected frequency), why a smaller sum of these values represent a smaller hydrologic alteration.

Response 6: We use the 25th and 75th percentiles as the defined RVA target range. But it is difficult to show the RVA target ranges of all the indicators.

Sorry for the mistake, the equation (28) has been modified to (|Observed frequency - Expected frequency|/Expected frequency).

Reviewer 3 Report

Well written paper with wide potential application.  However, a large aspect that this paper is missing is a section evaluating model performance and uncertainty at the various temporal scales.  On ln 516-517, authors conclude that “…considering the impact of hydrological uncertainty, calculation results from the model cannot directly guide the reservoir operation.  This indicates that uncertainty is a major issue that needs to be addressed and evaluated. 

Given that this paper is focused on an ecological dispatching model, the results and discussion section need to give a more thorough explanation of flow alteration impacts on the ecology.  How do the different approaches (i.e. POF, current scheduling, optimization model) impact flow alteration of the various flow metrics and what does that mean in terms of the biology?  This will give greater justification on the need of a multi-scale model.  Additionally, there was no results or discussion on the hydraulic metrics that were deemed important at the beginning of the paper. 

Specific comments:

Ln 143: How did you determine the reference period?  How many years of observed data was used and why?  Recommend moving Ln 423-424 to this section.

Ln 150-151: Was Ne(La) from the observed “reference” timeseries?  Make that clear.

Ln 202-203: Specific water depth will also differ depending on the local channel morphology.

Ln 214: Citation for Advanced Aquatic Biology?

Ln 220: What is the justification for using the species hydraulic requirements for Gymnocypris eckloni herzenstein?  Are they indicator species or species of special concern?  If their needs are the basis for your ecological targets, you need a sufficient explanation.

Ln 364: missing semicolon before 3).

Ln 365: What type of habitat change and how does that relate to changes in flow? 

Ln 370-371: Citations for the YRCC and Hydrological Yearbook?

Ln 376-382: Is this part of the caption for Figure 2? Reformat.

Ln 425: How is the ecological water demand estimated?

Ln 443: What do the colors mean in Figure 3?

Ln 452: Please expand on what Figure 4 shows in terms of iterations and optimal solutions.

Ln 495: Discussion needs to be expanded.  Need to describe the broader context of the findings.

Ln 496-498: Delete

Author Response

Thank you very much for your advice. Firstly the results of the small scale model provide more detailed and abundant information. Its only drawback is that it is difficult to solve. Thus we didn’t compare the performance and uncertainty at the various temporal scales. Then making the multi-scale coupled scheduling rules against uncertainty is a complicated work. In fact we have already started this part of work and achieved some preliminary results. We use more flexible decoupling method to optimize the scheduling rules directly. This model provides a framework for our future work. But in this paper we have only made some simple scheduling recommendations to deal with uncertainty. At last the ecological significances of characteristic flow regimes were described in the second chapter of this paper. However it is a pity that we don't have the conditions to get biological response data. This limits our assessment of the biological response to simulated flows. The hydraulic metrics that were mentioned at the beginning of the paper was converted to flow in the model. It is treated in small scale models because hydraulic parameters are sensitive to time scales. We expected that the duration of the simulated flow within a biologically appropriate range is not less than the duration of natural flow. Actually this is an easy goal to achieve. Thus we didn't show this result.

1. Ln 143: How did you determine the reference period?  How many years of observed data was used and why?  Recommend moving Ln 423-424 to this section.

Response 1: In this study, the daily flow from 1958 to 1977 was used as the reference series—as the values closely reflect natural runoff. 2007-2014 were selected as the scheduling period. We have moving Ln 423-424 to this section.

2. Ln 150-151: Was Ne(La) from the observed “reference” timeseries?  Make that clear.

Response 2: Ne(La) is were calculated according to the reference series. We have modified this part of content.

3. Ln 202-203: Specific water depth will also differ depending on the local channel morphology.

Response 3: We have modified this part of content.

4. Ln 214: Citation for Advanced Aquatic Biology?

Response 4: We have modified this part of content.

5. Ln 220: What is the justification for using the species hydraulic requirements for Gymnocypris eckloni herzenstein?  Are they indicator species or species of special concern?  If their needs are the basis for your ecological targets, you need a sufficient explanation.

Response 5: Gymnocypris eckloni herzenstein is very sensitive to water quality and hydrological regime. Thus it was chosen as keystone species. Many domestic scholars regard it as keystone species.[1]

6. Ln 364: missing semicolon before 3).

Response 6: Sorry for the mistake. We have modified this part of content.

7. Ln 365: What type of habitat change and how does that relate to changes in flow?

Response 7: It is wetted perimeter. The wetted perimeter varies with the flow.

8. Ln 370-371: Citations for the YRCC and Hydrological Yearbook?

Response 8: We have modified this part of content.

9. Ln 376-382: Is this part of the caption for Figure 2? Reformat.

Response 9: Yes. We have added the ’In the figure 2’ before this part.

10. Ln 425: How is the ecological water demand estimated?

Response 10: It is from the ecological flow requirements of the Chinese government for the Huayuankou section.

11. Ln 443: What do the colors mean in Figure 3?

Response 11: They respectively represent the degree of hydrological alteration of high, middle and low category.

12. Ln 452: Please expand on what Figure 4 shows in terms of iterations and optimal solutions.

Response 12: The large-scale model and small-scale model results can efficiently converge to the optimal solutions within 200 iterations.

Ln 495: Discussion needs to be expanded.  Need to describe the broader context of the findings.

Response 13: The multi-scale coupled ecological dispatching model provides a framework for the formulation of multi-scale scheduling rules. The more flexible decoupling method can be used by the model to optimize the scheduling rules directly. In further research, the large-scale model can optimize the operation rules, while the small-scale model can optimize the discharge process of reservoirs. We have added the above to the discussion section.

Ln 496-498: Delete

Response 14: Sorry for the mistake. We have deleted this part of content.

References:

[1].   M. Li, Determination of ecological water demand based on necessary flow depth and velocity for specific ecological function Journal of Hydraulic Engineering, 2007(06): P738-742.

Reviewer 4 Report

The manuscript by Tao Zhou et al. presents a modelling approach which allows scheduling of reservoir operation in such a way that hydrological alteration  is reduced. Due to the large extent of reservoirs worldwide and their influence on stream-flow regimes, this contribution is very interesting and relevant. However, the paper needs to be improved so that the results can be adequately evaluated and that conclusions can be drawn from this study to other regions. 

Most importantly, a quantitative assessment of the results obtained from current scheduling vs. alternative options is necessary - some results are presented in Fig. 3 & Fig. 6 but a thorough quantitative assessment is missing. Especially, this assessment should focus on ecological relevance - the importance of the flow pattern to a fish species is mentioned - but the implication of the results on this species are not evaluated.

Furthermore, a quantitative performance evaluation of model results vs. observations needs to be added - a presentation as in Fig. 5 is not sufficient.

The paper lacks a thorough discussion of the results especially in comparison with findings of the literature and a statement of the applicability of the method and the importance of the findings to other regions. Additionally, the aim of the research needs to be stated in the introduction. 

Specific comments:

Add numbers for references written in full - e.g. Francisco et al. (l. 64) and check that these appear in the reference list.

Table 1: only one species is presented - omit the species column to avoid confusion. Add the exact reference (author, year) for the data. In the table caption and the text: use the correct capitalization of the species and put it in italics. 

Figure 3 & 6: Needs larger bottom margins for readability of all descriptors. Legend missing. Dimension of values missing.

Line 369/371:: provide references

Line 469: Reference missing

Line 496-498: Is this a copy-paste error

Author Response

It is a pity that we don't have the conditions to get biological response data. This limits our assessment of the biological response to simulated flows. The hydraulic metrics that were mentioned at the beginning of the paper was converted to flow in the model. It is treated in small scale models because hydraulic parameters are sensitive to time scales. We expected that the duration of the simulated flow within a biologically appropriate range is not less than the duration of natural flow. Actually this is an easy goal to achieve. Thus we didn't show this result. Due to the lack of adequate biological response data, we still choose the degree of hydrologic alteration as the ecological evaluation index. This paper focuses on the construction of multi-scale model. This method is suitable for areas where the hydrological regime is seriously affected by dam operation. We will improve this in future studies. But at this stage, we still can't get biological response data to make more detailed evaluation. Additionally, the aim of the research has been added to the introduction.

1. Add numbers for references written in full - e.g. Francisco et al. (l. 64) and check that these appear in the reference list.

Response 1: Francisco’s full name is Francisco J. Peñas. Here's the reference:

[16] Peñas, F.J. and J. Barquín, Assessment of large-scale patterns of hydrological alteration caused by dams. Journal of Hydrology, 2019. 572, P 706-718.

We rechecked the references. Thank you for your advice.

2. Table 1: only one species is presented - omit the species column to avoid confusion. Add the exact reference (author, year) for the data. In the table caption and the text: use the correct capitalization of the species and put it in italics.

Response 2: We have omitted the species column, used the correct capitalization of the species and put it in italics.

3. Figure 3 & 6: Needs larger bottom margins for readability of all descriptors. Legend missing. Dimension of values missing.

Response 3: We redrew figure 3 and 6. Legends respectively represent the degree of hydrological alteration of high, middle and low category.

4. Line 369/371:: provide references

Response 4: We have added references to the hydrological yearbook and Comprehensive Planning of the Yellow River Basin.

5. Line 469: Reference missing

Response 5: We have added references to the POF.

6. Line 496-498: Is this a copy-paste error

Response 6: Sorry for the mistake. We have deleted this part of content.

Round 2

Reviewer 3 Report

Although the manuscript was improved by addressing the specific comments, I agree with Reviewer 4 in that the major issues presented by the reviewers in round 1 were not addressed.  Although biological response data is not available, the flow alteration results still need to be explained in terms of mechanisms that can impact the fish species presented in this paper.  I understand that this is a highly complicated model, but this alone gives even more reason to present model evaluation and an uncertainty discussion.   Please address the 6 points provided by Reviewer 4 (round 2).

Ln 364: “Legend” misspelled

Author Response

The manuscript has improved in the first round of revision, mainly by corrections following the specific comments raised by three reviewers.

However, the more important general comments on the manuscript have mostly not been addressed or responded to.

As a reminder, some of these (mentioned by various reviewers) were:

(1) addressing uncertainty

Response 1: To solve the problem of uncertainty, we use autoregressive model to generate 9 sets of inflow runoff series. Ten sets of inflow runoff series (Including the actual inflow series from 2007-2014) were used as input for simulation calculation. The final result is the average of 10 sets of calculated results.

(2) the results and discussion section need to give a more thorough explanation of flow alteration impacts on the ecology - i.e. - evaluating implication of the results on the fish species

Response 2: The quantitative evaluation and biological significance of various indicators under the POF approach and optimization model respectively were shown in Table 3. We found that the calculation results of the optimization model performed well in the second, third and fifth categories of indicators. That means the model performed better in terms of vegetation expansion, fluvial topography, natural habitat construction, fish migration and spawning and life cycle reproduction compared to current scheduling and the POF approach.

Table 3. Quantitative evaluation of the results.

Index

Current scheduling

POF approach

Optimization model

Ecosystem influences [39]

Magnitude of monthly discharge conditions

0.81

0.67

0.60

To meet the habitat   needs of aquatic organisms, the needs of plants for soil moisture content,   the water needs of terrestrial organisms with high reliability, the migration   needs of carnivores, and the influences of water temperature and oxygen   content.

Magnitude and duration of annual extreme discharge conditions

0.72

0.88

0.41

To meet the needs of   vegetation expansion, river topography and natural habitat construction,   nutrient exchange in rivers and flood detention areas, distribution of plant   communities in lakes, ponds and flood detention areas.

Timing of annual extreme discharge conditions

0.99

1.22

0.27

To meet the migration of   fish spawning, the cycle of life reproduction, biological breeding habitat   conditions, species evolution needs.

Frequency and duration of high/low flow pulses

0.83

0.71

0.70

To generate the frequency   and magnitude of soil moisture stress for plants.

Rate/frequency of hydrograph changes

0.71

1.12

0.67

To meet the drought of   plants, the trapping of organics on the island and in the flood detention   area, and the drying stress of low-speed organisms.

Suitable ecological velocity

0.29

0.40

0.46

To meet the appropriate   velocity requirements of fish.

  Table 3. shows the quantitative evaluation and biological significance of various indicators under the POF approach and optimization model. The values of the first five categories of indicators are degree of hydrologic alterations. The value of the last indicator represents the proportion of the number of days in the suitable ecological velocity range to the total number of days.

The above has been added into the Line 517.

(3)  discussion on the hydraulic metrics

Response 3: From the perspective of hydraulic parameters, the number of days within the appropriate velocity range during the reference series (1958-1977) is 2117, which is 29% of the total number of days. The number of days within the appropriate velocity range during the scheduling period (2007-2014) is 1452, which is 50% of the total number of days. Due to the influence of flow flattening, the number of days suitable for fish has increased. We expected that the duration of the simulated flow within the appropriate velocity is not less than the duration of natural flow. The number of days within the appropriate velocity range in the simulated flow accounted for 46% of the total. And the number of days within the appropriate velocity range under POF approach accounted for 43% of the total. Obviously, this is an easy goal to achieve.

The above has been added into the Line 508.

(4) a quantitative assessment of the results obtained from current scheduling vs. alternative options

Response 4: The quantitative assessment of the results obtained from current scheduling vs. alternative options was shown in Table 3. We list the values of various indicators and the corresponding ecological significance in table 3.

 (5) thorough discussion of the results especially in comparison with findings of the literature and a statement of the applicability of the method and the importance of the findings to other regions

Response 5: Compared with existing research, the existing model either directly optimizes the scheduling rules [17] or evaluates the existing daily flow process [24]. However, the rule optimization model cannot describe the small scale runoff process in detail and the runoff evaluation model cannot directly guide reservoir operation. Therefore a modelling approach which allows scheduling of reservoir operation in such a way that hydrological alteration is reduced was presented in this work. The multi-scale coupled ecological dispatching model provides a framework for the formulation of multi-scale scheduling rules. The more flexible decoupling method can be used by the model to optimize the scheduling rules directly. In further research, the large-scale model can optimize the operation rules, while the small-scale model can optimize the discharge process of reservoirs. This method is suitable for areas where the hydrological regime is seriously affected by dam operation. This model is more suitable for large rivers because of its attention to the degree of hydrological alterations. For other river basins, decision makers can choose the key ecological issues of the river basin. And the weight of the objective function can be formulated according to the key ecological problems and the ecological significance of various indicators. Scheduling rules suitable for this basin can be found by this way.

The above has been added into the Line 551.

(6) the aim of the research needs to be stated in the introduction

Response 6: The multi-scale coupled model also provides a framework for the formulation of multi-scale scheduling rules. This model can help decision-makers to determine whether there is still room for improvement in ecological scheduling. It also provides decision-makers with more information to look for ways to improve.

The above has been added into the introduction.  Line 113

(7)Ln 364: “Legend” misspelled

Response 7: Figures has been modified.

Thank you very much for your advice.

These points need to be addressed in the manuscript

Reviewer 4 Report

The manuscript has improved in the first round of revision, mainly by corrections following the specific comments raised by three reviewers. 

However, the more important general comments on the manuscript have mostly not been addressed or responded to. 

As a reminder, some of these (mentioned by various reviewers) were:

(1) addressing uncertainty

(2) the results and discussion section need to give a more thorough explanation of flow alteration impacts on the ecology - i.e. - evaluating implication of the results on the fish species

(3)  discussion on the hydraulic metrics 

(4) a quantitative assessment of the results obtained from current scheduling vs. alternative options

(5) thorough discussion of the results especially in comparison with findings of the literature and a statement of the applicability of the method and the importance of the findings to other regions

(6) the aim of the research needs to be stated in the introduction

These points need to be addressed in the manuscript.

Author Response

The manuscript has improved in the first round of revision, mainly by corrections following the specific comments raised by three reviewers.

However, the more important general comments on the manuscript have mostly not been addressed or responded to.

As a reminder, some of these (mentioned by various reviewers) were:

(1) addressing uncertainty

Response 1: To solve the problem of uncertainty, we use autoregressive model to generate 9 sets of inflow runoff series. Ten sets of inflow runoff series (Including the actual inflow series from 2007-2014) were used as input for simulation calculation. The final result is the average of 10 sets of calculated results.

(2) the results and discussion section need to give a more thorough explanation of flow alteration impacts on the ecology - i.e. - evaluating implication of the results on the fish species

Response 2: The quantitative evaluation and biological significance of various indicators under the POF approach and optimization model respectively were shown in Table 3. We found that the calculation results of the optimization model performed well in the second, third and fifth categories of indicators. That means the model performed better in terms of vegetation expansion, fluvial topography, natural habitat construction, fish migration and spawning and life cycle reproduction compared to current scheduling and the POF approach.

Table 3. Quantitative evaluation of the results.

Index

Current scheduling

POF approach

Optimization model

Ecosystem influences [39]

Magnitude of monthly discharge conditions

0.81

0.67

0.60

To meet the habitat   needs of aquatic organisms, the needs of plants for soil moisture content,   the water needs of terrestrial organisms with high reliability, the migration   needs of carnivores, and the influences of water temperature and oxygen   content.

Magnitude and duration of annual extreme discharge conditions

0.72

0.88

0.41

To meet the needs of   vegetation expansion, river topography and natural habitat construction,   nutrient exchange in rivers and flood detention areas, distribution of plant   communities in lakes, ponds and flood detention areas.

Timing of annual extreme discharge conditions

0.99

1.22

0.27

To meet the migration of   fish spawning, the cycle of life reproduction, biological breeding habitat   conditions, species evolution needs.

Frequency and duration of high/low flow pulses

0.83

0.71

0.70

To generate the frequency   and magnitude of soil moisture stress for plants.

Rate/frequency of hydrograph changes

0.71

1.12

0.67

To meet the drought of   plants, the trapping of organics on the island and in the flood detention   area, and the drying stress of low-speed organisms.

Suitable ecological velocity

0.29

0.40

0.46

To meet the appropriate   velocity requirements of fish.

 Table 3. shows the quantitative evaluation and biological significance of various indicators under the POF approach and optimization model. The values of the first five categories of indicators are degree of hydrologic alterations. The value of the last indicator represents the proportion of the number of days in the suitable ecological velocity range to the total number of days.

The above has been added into the Line 517.

(3)  discussion on the hydraulic metrics

Response 3: From the perspective of hydraulic parameters, the number of days within the appropriate velocity range during the reference series (1958-1977) is 2117, which is 29% of the total number of days. The number of days within the appropriate velocity range during the scheduling period (2007-2014) is 1452, which is 50% of the total number of days. Due to the influence of flow flattening, the number of days suitable for fish has increased. We expected that the duration of the simulated flow within the appropriate velocity is not less than the duration of natural flow. The number of days within the appropriate velocity range in the simulated flow accounted for 46% of the total. And the number of days within the appropriate velocity range under POF approach accounted for 43% of the total. Obviously, this is an easy goal to achieve.

The above has been added into the Line 508.

(4) a quantitative assessment of the results obtained from current scheduling vs. alternative options

Response 4: The quantitative assessment of the results obtained from current scheduling vs. alternative options was shown in Table 3. We list the values of various indicators and the corresponding ecological significance in table 3.

 (5) thorough discussion of the results especially in comparison with findings of the literature and a statement of the applicability of the method and the importance of the findings to other regions

Response 5: Compared with existing research, the existing model either directly optimizes the scheduling rules [17] or evaluates the existing daily flow process [24]. However, the rule optimization model cannot describe the small scale runoff process in detail and the runoff evaluation model cannot directly guide reservoir operation. Therefore a modelling approach which allows scheduling of reservoir operation in such a way that hydrological alteration is reduced was presented in this work. The multi-scale coupled ecological dispatching model provides a framework for the formulation of multi-scale scheduling rules. The more flexible decoupling method can be used by the model to optimize the scheduling rules directly. In further research, the large-scale model can optimize the operation rules, while the small-scale model can optimize the discharge process of reservoirs. This method is suitable for areas where the hydrological regime is seriously affected by dam operation. This model is more suitable for large rivers because of its attention to the degree of hydrological alterations. For other river basins, decision makers can choose the key ecological issues of the river basin. And the weight of the objective function can be formulated according to the key ecological problems and the ecological significance of various indicators. Scheduling rules suitable for this basin can be found by this way.

The above has been added into the Line 551.

(6) the aim of the research needs to be stated in the introduction

Response 6: The multi-scale coupled model also provides a framework for the formulation of multi-scale scheduling rules. This model can help decision-makers to determine whether there is still room for improvement in ecological scheduling. It also provides decision-makers with more information to look for ways to improve.

The above has been added into the introduction.  Line 113

These points need to be addressed in the manuscript

Round 3

Reviewer 3 Report

Manuscript was significantly improved by the changes.  However, there needs to be an adequate discussion on the uncertainty aside from just running the 10 additional simulations and calculating the average values.  Please see list below for additional feedback:

Ln 116-117: reword sentence “It also provides decision-makers…” Sounds repetitive to the previous sentence.

Ln 367: “Legeng” needs to be “Legend”

Ln 519: The average does not tell you much about uncertainty.  Please also list the range of the current scheduling, POF, and optimization for each of the indices in Table 3.  If the range of alteration overlaps between the 3 categories, then your confidence in comparing outcomes in that category would be lower.  If the ranges do not overlap, then confidence in results should be higher for a given index.  Please discuss your results along these lines and provide a discussion on the uncertainty rather than just listing average values. 

Author Response

Manuscript was significantly improved by the changes.  However, there needs to be an adequate discussion on the uncertainty aside from just running the 10 additional simulations and calculating the average values.  Please see list below for additional feedback:

Ln 116-117: reword sentence “It also provides decision-makers…” Sounds repetitive to the previous sentence.

Response 1: This sentence has been modified to:

This model can help decision-makers to determine whether there is still room for improvement in ecological scheduling and look for ways to improve.

Ln 367: “Legeng” needs to be “Legend”

Response 2: Sorry for this mistake, we have corrected the figure 1.

Ln 519: The average does not tell you much about uncertainty.  Please also list the range of the current scheduling, POF, and optimization for each of the indices in Table 3.  If the range of alteration overlaps between the 3 categories, then your confidence in comparing outcomes in that category would be lower.  If the ranges do not overlap, then confidence in results should be higher for a given index.  Please discuss your results along these lines and provide a discussion on the uncertainty rather than just listing average values. The corresponding parts are modified as follows:

Response 3: We have listed the range of the POF, and optimization for each of the indices in Table 3. As it is difficult to simulate decision-makers' decisions, the uncertainty of current scheduling is not shown. The hydrological alterations of the current scheduling are only used as a reference in table 3.

To solve the problem of uncertainty, we use autoregressive model to generate 9 sets of inflow runoff series. Ten sets of inflow runoff series (Including the actual inflow series from 2007-2014) were used as input for simulation calculation. The final result is the average of 10 sets of calculated results and the variation range of 10 sets of results. The values above are the mean, and the values in brackets are the variation ranges. The quantitative evaluation and biological significance of various indicators under the POF approach and optimization model respectively were shown in Table 3. As it is difficult to simulate decision-makers' decisions, the uncertainty of current scheduling is not shown. The hydrological alterations of the current scheduling are only used as a reference in table 3. We found that the calculation results of the optimization model performed well in the second, third and fifth categories of indicators. That means the model performed better in terms of vegetation expansion, fluvial topography, natural habitat construction, fish migration and spawning and life cycle reproduction compared to current scheduling and the POF approach. In terms of uncertainty, the optimization model performs better than the POF approach too. Especially in the second, third and fifth categories of indicators, the performance of the optimization model are far better than the POF approach. And the variation ranges barely overlap. That means the model performed better in addressing uncertain problems compared to current scheduling and the POF approach in terms of the above ecological problems.

Table 3. Quantitative evaluation of the results.

Index

Current scheduling

POF approach

Optimization model

Ecosystem influences   [39]

Magnitude of monthly discharge conditions

0.81

0.67

(0.62-0.71)

0.60

(0.55-0.63)

To meet the habitat   needs of aquatic organisms, the needs of plants for soil moisture content,   the water needs of terrestrial organisms with high reliability, the migration   needs of carnivores, and the influences of water temperature and oxygen   content.

Magnitude and duration of annual extreme discharge conditions

0.72

0.88

(0.80-0.95)

0.41

(0.40-0.42)

To meet the needs of   vegetation expansion, river topography and natural habitat construction,   nutrient exchange in rivers and flood detention areas, distribution of plant   communities in lakes, ponds and flood detention areas.

Timing of annual extreme discharge conditions

0.99

1.22

(0.88-1.44)

0.27

(0.25-0.29)

To meet the migration of   fish spawning, the cycle of life reproduction, biological breeding habitat   conditions, species evolution needs.

Frequency and duration of high/low flow pulses

0.83

0.71

(0.68-0.75)

0.70

(0.69-0.72)

To generate the frequency   and magnitude of soil moisture stress for plants.

Rate/frequency of hydrograph changes

0.71

1.12

(0.99-1.20)

0.67

(0.65-0.69)

To meet the drought of   plants, the trapping of organics on the island and in the flood detention   area, and the drying stress of low-speed organisms.

Suitable ecological velocity

0.29

0.40

(0.38-0.42)

0.46

(0.43-0.49)

To meet the appropriate   velocity requirements of fish.

Table 3. shows the quantitative evaluation and biological significance of various indicators under the POF approach and optimization model. The values of the first five categories of indicators are degree of hydrologic alterations. The value of the last indicator represents the proportion of the number of days in the suitable ecological velocity range to the total number of days.

Reviewer 4 Report

The manuscript improved substantially during the revision process and I can now recommend publication. Minor comment: "Legend" is still wrongly spelled as "Legeng" in figure 1 - this should be corrected. 

Author Response

The manuscript improved substantially during the revision process and I can now recommend publication. Minor comment: "Legend" is still wrongly spelled as "Legeng" in figure 1 - this should be corrected.

Response: Sorry for this mistake, we have corrected the figure 1.

Round 4

Reviewer 3 Report

Sufficient improvement to the manuscript with the minor changes.  Accept for publication.